# Effects of Pharmacological Thermogenic Adipocyte Activation on Metabolism and Atherosclerotic Plaque Regression

**DOI:** 10.3390/nu11020463

**Published:** 2019-02-23

**Authors:** Anna Worthmann, Christian Schlein, Jimmy F. P. Berbée, Patrick C. N. Rensen, Joerg Heeren, Alexander Bartelt

**Affiliations:** 1Department of Biochemistry and Molecular Cell Biology, University Medical Center Hamburg-Eppendorf, 20246 Hamburg, Germany; a.worthmann@uke.de (A.W.); c.schlein@uke.de (C.S.); heeren@uke.de (J.H.); 2Institute of Human Genetics, University Medical Center Hamburg-Eppendorf, 20246 Hamburg, Germany; 3I. Department of Internal Medicine, University Medical Center, Hamburg-Eppendorf, 20246 Hamburg, Germany; 4Department of Medicine, Division of Endocrinology and Einthoven Laboratory for Experimental Vascular Medicine, Leiden University Medical Center, 2300 RC Leiden, The Netherlands; J.F.P.Berbee@lumc.nl (J.F.P.B.); P.C.N.Rensen@lumc.nl (P.C.N.R.); 5Institute for Cardiovascular Prevention (IPEK), Ludwig-Maximilians-University, 81377 Munich, Germany; 6German Center for Cardiovascular Research (DZHK), Partner Site Munich Heart Alliance, 80336 Munich, Germany

**Keywords:** brown adipose tissue, browning, thermogenesis, cholesterol, triglyceride, atherosclerosis, LDLR

## Abstract

Thermogenic adipocytes burn nutrients in order to produce heat. Upon activation, brown adipose tissue (BAT) clears vast amounts of lipids and glucose from the circulation and thus substantially lowers plasma lipid levels. As a consequence, BAT activation protects from the development of atherosclerosis. However, it is unclear if pharmacologic activation of BAT can be exploited therapeutically to reduce plaque burden in established atherosclerotic disease. Here we study the impact of thermogenic adipose tissues on plaque regression in a mouse model of atherosclerosis. Thermogenic adipocytes in atherosclerotic low-density lipoprotein (LDL) receptor (LDLR)-deficient mice were pharmacologically activated by dietary CL316,243 (CL) treatment for 4 weeks and the outcomes on metabolically active tissues, plasma lipids and atherosclerosis were analyzed. While the chronic activation of thermogenic adipocytes reduced adiposity, increased browning of white adipose tissue (WAT), altered liver gene expression, and reduced plasma triglyceride levels, atherosclerotic plaque burden remained unchanged. Our findings suggest that despite improving adiposity and plasma triglycerides, pharmacologic activation of thermogenic adipocytes is not able to reverse atherosclerosis in LDLR-deficient mice.

## 1. Introduction

Cardiovascular diseases (CVD) represent a critical threat to human health and account for roughly 31.3% of global mortality [1]. Dyslipidemia and in particular hypercholesterolemia are major risk factors for the development of atherosclerosis. A promising approach to counteract dyslipidemia and reduce hypercholesterolemia is the activation of thermogenic adipocytes [2,3]. These specialized fat cells utilize high-energy nutrients in order to produce heat and thus defend the body against cold stress. Physiologically, thermogenic adipocytes are activated upon a cold stimulus, which results in the release of norepinephrine from sympathetic nerve terminals and its subsequent binding to adrenergic receptors on adipocytes. This physiological activation can be mimicked pharmacologically by treatment with the selective β_3_-adrenergic receptor agonist CL316,243 (CL) [4,5]. When activated, the presence of a unique protein in thermogenic adipocytes, the uncoupling protein 1 (UCP1), enables exothermic uncoupling of the respiratory chain and thus promotes a process known as adaptive thermogenesis [5]. Brown and also beige adipocytes (i.e., brown-like adipocytes arising in WAT after prolonged cold-exposure or upon chronic pharmacologic activation in a process termed browning) are highly metabolically active [6]. Upon activation, intravascular hydrolysis of triglyceride-rich lipoproteins (TRL) results in the release and subsequent utilization of free fatty acids (FFA) by adipocytes as well as enhanced flux of cholesterol-enriched remnant particles to the liver, where they are cleared from the circulation [3]. Thus, thermogenic adipocytes substantially reduce plasma triglyceride and cholesterol levels. Furthermore, activation of BAT also promotes HDL-mediated reverse cholesterol transport to the liver [7] as well as the excretion of cholesterol in terms of bile acids [8], altogether contributing to an anti-atherosclerotic metabolic phenotype. Active thermogenic adipose tissue is also present in adult humans [9,10,11,12,13,14] and its presence is inversely correlated with body mass index (BMI) [15,16]. In humans, BAT can be recruited by cold acclimation [17] and enhances energy expenditure [18,19].

The combined effects of acute reduction of plasma lipids as well as long-term reduction in adiposity make the activation of thermogenic adipocytes an attractive target for the treatment of atherosclerosis. The effects of cold-mediated or pharmacological activation of thermogenic adipocytes on the progression of atherosclerosis have already been explored [3,20,21] but to date, no study investigated the therapeutic potential of BAT activation for the treatment of established atherosclerosis and atherosclerosis regression. Regression of atherosclerosis refers to the stabilization of plaques due to reduced lipid and enhanced fibrous content of lesions [22] and was described already in the 1920s [23]. While wild-type mice do not develop atherosclerosis, over the last decades the development of transgenic technology has made it possible to study atherosclerosis in preclinical mouse models. To date, many studies have underlined the importance of plasma lipid lowering for regression of atherosclerosis [24,25,26,27]. Contrary to other mouse models of atherosclerosis, low-density lipoprotein receptor (LDLR)-deficient mice phenocopy human patients with familial hypocholesteremia [28], show a rather human-like pro-atherogenic lipid profile (high low-density lipoprotein (LDL)) and due to the presence of functional apolipoprotein E (APOE), the development of concomitant inflammation in atherosclerosis is not compromised (unlike in APOE-deficient mice) [29,30]. Interestingly, regression of atherosclerosis in LDLR-deficient mice has also been shown to occur with longer periods of lipid lowering solely induced by dietary intervention [31].

Here we investigate the effects of pharmacological activation of thermogenic adipocytes on the regression of established atherosclerosis in LDLR-deficient mice. Therefore, we designed dietary regimen switching high-fat, high-cholesterol, and high-sucrose-diet (HFCS)-fed LDLR-deficient mice to regular chow diet or to chow diet supplemented with CL316,243 (CL). We monitored body weight, plasma lipid levels and analyzed liver, BAT, and browning of WAT as well as the severity of atherosclerotic lesions.

## 2. Materials and Methods

### 2.1. Experimental Animals, Housing Conditions, Diets, and Animal Experiments

All animal experiments were approved by the Animal Welfare Officers of University Medical Center Hamburg-Eppendorf (UKE) and Behörde für Gesundheit und Verbraucherschutz Hamburg. LDLR-deficient mice [32] were bred and housed in the animal facility of UKE at 22 °C with a day-night cycle of 12 h with *ad libitum* access to food and water. For the experiments, male 8 week-old LDLR-deficient mice were fed a HFCS (Ssniff S8301-E020; 21% butter, 0.2% cholesterol, 35.5% sucrose) for 12 weeks followed by feeding of a standard chow diet (Lasvendi, Rod16R) (Mock) or supplemented with 5 mg CL316,243 (Cayman) (CL) pro kg chow diet. To generate the CL-containing diet, CL was first dissolved in Saline at 1 mg/mL, then diluted 1:10 in EtOH, mixed with the chow diet and incubated over night until the EtOH was evaporated. After switching to chow diet, mice were housed in single cages, food intake was measured daily, and body weight was monitored weekly. Blood samples for plasma lipid analysis were withdrawn from the tail vein of 4 h fasted mice at indicated time points. Tissue and blood collections were performed after a 4 h fasting period. Mice were anesthetized with a lethal dose (15 µL/g mouse bodyweight) of a mixture containing Ketamin (25 mg/mL)/Xylazin (0.2%) in 0.9% NaCl. Blood was withdrawn by cardiac puncture with syringes containing 5 µL 0.5 M EDTA for plasma preparation. Animals were perfused with 5 mL ice-cold PBS containing 10 µ/mL heparin. Organs were harvested and immediately conserved either in TriFast™ (Peqlab, Erlangen, Germany) for RNA analysis, in 3.7% formaldehyde solution for histology or snap-frozen in liquid nitrogen and stored at −80 °C for further processing. Hearts with the attached aortae were stored in 3.7% formaldehyde solution.

### 2.2. Plasma Analysis

Plasma was generated by centrifugation of EDTA-spiked blood for 10 min at 10,000 rpm at 4 C in a bench top centrifuge. Plasma cholesterol and triglycerides were determined using commercial kits (Roche) that were adapted to 96-well microtiter plates. Precipath^®^ (Roche, Mannheim, Germany) was used as a standard for cholesterol as well as triglycerides. For lipoprotein profiling 200 µL of pooled plasma was separated by fast-performance liquid chromatography (FPLC) on a Superose™ 6 10/300 GL column (GE Healthcare, Freiburg, Germany) with a flow rate of 0.5 mL/min. 30 Fractions (volume of faction 0.5 mL) were collected. Triglycerides as well as cholesterol concentrations were measured in each fraction. 

### 2.3. Gene Expression Analysis

After the disruption of tissue samples in TriFast™ (Peqlab, Erlangen, Germany) using a Qiagen Tissue Lyzer, nucleic acids were extracted with chloroform before RNA was purified using RNA Purification Kit NucleoSpin^®^ RNA II (Macherey-Nagel, Düren, Germany) following the manufacturer’s instructions. By means of SuperScript^®^ III Reverse Transcriptase (Thermo Fisher Scientific, Darmstadt, Germany) synthesis of complementary DNA was performed. Quantitative real-time PCR reactions for indicated genes were conducted on a 7900HT sequence detection system using TaqManAssay-on-Demand primer sets (Thermo Fisher Scientific, Darmstadt, Germany, mAbcg5: Mm00446249_m1, mAbcg8: Mm00445970_m1, mCyp7a1: Mm00484150_m1, mElovl3: Mm00468164_m1, mFasn: Mm00662319_m1, mLpl: Mm00434764_m1, mPpara: Mm00440939_m1, mPpargc1a: Mm00447183_m1, mUcp1: Mm00494069_m1). Cycle thresholds (Cts) were normalized to TATA-box binding protein (*Tbp*) house keeper levels by using the ΔΔCt method.

### 2.4. Histology

In 3.7% formaldehyde solution fixed organs were embedded in paraffin, cut and hematoxylin and eosin (HE) stains were performed using standard protocols as described previously [33].

### 2.5. Liver Analysis

In liver homogenates, protein concentration was measured according to the method of Lowry [34]. Liver triglyceride levels were quantified using a commercial kit (Roche) that was adapted to 96-well microtiter plates with Precipath^®^ (Roche, Mannheim, Germany) as standard as described previously [35]. Liver cholesterol levels were determined using Amplex^®^ Red Cholesterol Assay Kit (Thermo Fisher Scientific, Darmstadt, Germany) as described previously [36].

### 2.6. Analysis of Atherosclerosis

For *en face* analysis, aorta-surrounding adventitial fatty tissue was removed carefully, aortae were opened longitudinally and pinned with Austerlitz^®^ insect pins (Entomoravia, Slavkov u Brna, Czech Republic) on a wax plate. Aortae were rinsed in 60% isopropanol for 1 min and then stained in Sudan IV solution (1 mg/mL in 60% isopropanol) for 10 min. Destaining was performed in 60% isopropanol and aortae were stored in 3.7% formaldehyde solution until pictures were taken. Quantification of plaque size was performed with ImageJ. For aortic root staining, hearts were fixed in 3.7% formaldehyde solution for 24 h, transferred into 70% EtOH and subsequently embedded in paraffin. Aortic root sectioning and staining as well as classification of lesion severity was performed as previously described [3]. 

### 2.7. Statistical Analysis

Data are expressed as mean ± S.E.M. GraphPad Prism 7.0 was used for statistical calculations and two-tailed, independent Student’s *t* test was assessed to compare differences between groups. Differences were considered as significant at a probability level (*p*) of 0.05 with: * *p* < 0.05, ** *p* < 0.01 and *** *p* < 0.001.

## 3. Results

### 3.1. CL Treatment Reduces Adiposity Irrespective of Food Intake

While the activation of thermogenic adipocytes has been shown to reduce hypercholesteremia and atherosclerosis development [3], the therapeutic effects of their activation on established lesions remain unclear. To study the effect of thermogenic adipocytes on atherosclerosis regression, 12 weeks HFCS-fed atherosclerotic LDLR-deficient mice were switched to chow diet or to chow diet supplemented with CL316,243 (CL) for 4 weeks to pharmacologically activate thermogenic adipocytes (Figure 1A).

During the time course of 4 weeks, food intake was measured daily, and body weight was monitored weekly. Daily food intake was not altered in CL-treated LDLR-deficient mice compared to control mice (Figure 1B) if at all slightly higher as indicated by cumulative food intake over time (Figure 1C). Dietary intervention reduced body weight in both groups by roughly 7 g (Figure 1D) and a substantial weight loss was already present during the first 3 days in both groups (Figure 1E). However, mice receiving CL-containing diet lost significantly more weight than mock diet-fed mice (Figure 1E). In line, the enhanced weight loss induced by CL treatment corresponded to reduced adiposity. Thus, CL-treatment enhanced weight loss independently of food intake. Liver and heart weights remained unaffected by the activation of thermogenic adipose tissues, whereas epidydimal white adipose tissue (epiWAT) weight was significantly lower and BAT and inguinal WAT (ingWAT) weights trended to be decreased in the CL-treated group (Figure 1F). These results indicate enhanced energy expenditure in CL-treated LDLR-deficient.

### 3.2. CL Treatment Activates Thermogenic Fat Depots 

Chronic treatment with CL results in chronic thermogenic activation and profound adipose tissue browning [37] In order to verify the pharmacologic activation of thermogenesis by CL treatment, we analyzed BAT and ingWAT of mock and CL-treated LDLR-deficient mice. Macroscopically, BAT depots of CL-treated LDLR-deficient mice appeared dark brown while BATs in mock-treated LDLR-deficient mice looked rather pale brown. Additionally, the ingWAT depots of mock-treated mice remained white, while the ingWAT of CL-treated LDLR-deficient mice appeared darker in color (Figure 2A), indicating ingWAT browning. 

In line with the macroscopic appearance, histology confirmed the marked change in tissue morphology. In both, BAT and ingWAT, CL treatment reduced the amount of lipids. BAT and ingWAT of CL-treated LDLR-deficient mice showed reduced lipid droplet size compared to chow controls (Figure 2B). Gene expression analysis of ingWAT further indicated the successful activation of thermogenic adipocytes, as the expression of surrogate markers of non-shivering thermogenesis such as *Ucp1* encoding UCP1, which facilitates proton leak, *Ppara*, a transcription factor promoting energy expenditure, and *Ppargc1a*, the major transcription factor of mitochondrial biogenesis, were significantly increased in ingWAT of CL-treated LDLR-deficient mice compared to mock-treated mice (Figure 2C) *Lpl* encodes lipoprotein lipase, which catabolizes triglycerides from lipoproteins and thus is a gate keeper for the uptake of non-esterified fatty acids into adipocytes [2,38]. *Elovl3* encodes a fatty-acid elongase, which is a marker of de novo lipogenesis [39]. Both *Elovl3* (by trend) and *Lpl* (significantly) were increased after CL treatment (Figure 2C). Altogether, these results indicate that after CL treatment, ingWAT is metabolically highly active, taking up lipids from the circulation as well as producing lipids de novo, likely to replenish thermogenic fuel stores.

Next, we wanted to investigate if the activation of thermogenic adipose tissues by CL treatment also affects other organs. The liver has a pivotal role in lipid metabolism, hence we analyzed the livers of mock- and CL-treated LDLR-deficient mice. Histologically, the livers remained largely unaffected by CL-treatment (Figure 2D), even though CL-treated mice displayed slightly less lipid deposition compared to mock-treated controls. This slight decrease was also mirrored in biochemically determined liver lipid levels, as triglyceride levels were reduced by trend (*p* < 0.1) in CL-treated LDLR-deficient mice. However, no such reduction was observed in liver cholesterol levels (Figure 2E). Interestingly, hepatic expression of genes involved in lipid metabolism was affected by pharmacologic activation of thermogenic adipose tissues. In line with the slightly decreased liver triglyceride levels, CL treatment resulted in modestly reduced expression of the gene encoding the fatty acid synthase, *Fasn*, the expression of which correlates with steatosis [40]. While the expression of cholesterol excretion transporters ATP binding cassette subfamily G members 5 and 8 (encoded by *Abcg5* and *Abcg8*) did not differ between CL- and mock-treated LDLR-deficient mice, CL induced the expression of *Cyp7a1* (Figure 2F). As CYP7A1 is the rate limiting hydroxylase responsible for the conversion of cholesterol to bile acids, the observed results are in line with previous reports that the activation of thermogenic adipocytes by cold or CL enhances the conversion of cholesterol into bile acids for excretion [8].

### 3.3. Decreased Plasma Triglycerides after CL Treatment Do Not Translate into Reduced Atherosclerosis

Activation of thermogenic adipose tissue is well known to impact on plasma triglyceride and cholesterol levels [2,3,7]. Thus, we analyzed the plasma before, during and after 4 weeks of chow-feeding supplemented with or without CL. Already after 1 week of chow-feeding, plasma cholesterol levels dropped roughly by half irrespective of CL treatment (Figure 3A). Over time, this reduction was further enhanced in both mock- and CL-treated LDLR-deficient mice. Interestingly, 4 weeks after the dietary intervention, CL treatment caused modest, albeit non-significant reductions of plasma cholesterol levels compared to control mice (Figure 3A). In agreement, also plasma triglyceride levels initially declined in response to the dietary intervention in both groups. However, the lipid lowering effect of CL treatment was more pronounced: Compared to control mice, which had similar plasma triglyceride levels in week 4 compared to week 0 (Figure 3A), CL-treated LDLR-deficient mice displayed diminished plasma triglyceride 2 weeks after starting the CL treatment (Figure 3A). The reductions of plasma triglyceride and cholesterol levels were confirmed in the lipoprotein profile determined by fast performance liquid chromatography (FPLC). Compared to the beginning of chow feeding (d0), cholesterol levels in the triglyceride-rich lipoprotein (TRL), low density lipoprotein (LDL) and high-density lipoprotein (HDL) fraction were substantially reduced in both mock- and CL-treated LDLR-deficient mice (Figure 3B). Furthermore, TRL cholesterol was lower in the CL-treated group, which is in line with the slight decrease in total plasma cholesterol levels (Figure 3B). Corresponding to total plasma triglyceride levels, the triglyceride content of the TRL fraction was only diminished in the CL-treated LDLR-deficient mice (Figure 3C). 

Next, we determined whether the beneficial effects of CL treatment in LDLR-deficient mice (accelerated body weight loss, reduced plasma lipids, and altered liver gene expression) would also translate into regression of atherosclerosis. We analyzed aortic plaque burden in mock- or CL-treated LDLR-deficient mice by *en face* staining of neutral lipids in the total aortae and also by analysis of the aortic roots. Macroscopically (Figure 3D) and by quantification (Figure 3E), we did not find any differences in lesion area between CL-treated and control mice neither in the total aorta, the abdominal aorta nor in the aortic arch. In agreement, CL treatment did also not alter lesion area in the aortic roots (Figure 3F). Also, lesion severity remained unaffected by CL treatment (Figure 3G). 

In conclusion, despite improving metabolic parameters such as body weight and plasma lipid levels, activation of thermogenic adipocytes by 4 weeks of dietary CL treatment did not improve atherosclerosis in LDLR-deficient mice.

## 4. Discussion

In light of the presence of thermogenic adipocytes in humans [9,10,11,12,13,14], the beneficial outcome of their activation on metabolic health, in particular their lipid lowering effects [2,3,7,8], these adipocytes have emerged as a promising target for the treatment of atherosclerosis. Interestingly, studies investigating the outcome of BAT activation on atherosclerosis have come to diverging conclusions, depending on the experimental approach. While wild-type mice respond to the plasma lipid lowering effects of cold or CL, they do not develop atherosclerosis, and therefore transgenic models have to be used to study this process. In response to cold, Dong et al. found increased plasma cholesterol levels due to accumulation either of LDL (in LDLR-deficient mice) or of TRL-remnant particles (in APOE-deficient mice) and subsequently increased atherosclerotic plaque development [20]. On the other hand, we found that the activation of thermogenic adipocytes by either cold or CL treatment reduced hypercholesterolemia and protected from the development of atherosclerosis in APOE3L.CETP- mice (E3L.CETP) [3]. In contrast to these studies, here, we rather focus on the question, if activation of BAT can be exploited therapeutically to induce regression of established atherosclerotic lesions. In line with previous studies [18,41], we found that BAT activation promoted body weight loss (Figure 1C) and reduced adiposity (Figure 1D). These body weight-lowering effects of CL were mediated by an increased energy expenditure due to higher activity of not only BAT but also ingWAT (Figure 2A). As described [37], but also as observed in this current study, CL not only stimulated activity of BAT, as visible by reduced lipid content in histology (Figure 2B) but also stimulated browning of WAT (Figure 2B,C). Interestingly we also found evidence, that CL treatment alters lipid metabolism in the liver. This is in line with previous reports, as on the one hand it has already been described, that BAT activation impacts on reverse cholesterol transport and thus influences liver cholesterol handling [7]. On the other hand, we found increased transcript levels of *Cyp7a1*, which is involved in the conversion of cholesterol into bile acids (Figure 2F). An enhanced conversion of cholesterol into bile acids in response to cold was also described by our group before [8] and is thought to be protective from systemic cholesterol overload. It is well established that BAT activation normalizes hypertriglyceridemia [2] and in the present study, the activation of BAT by CL in LDLR-deficient mice resulted in a marked reduction of plasma triglyceride levels (Figure 3A,C). This is also in line with the results of Dong et al. [20]. However, contrary to them we found plasma cholesterol levels to be non-significantly reduced rather than increased after CL treatment, irrespectively of the diet (Figure 3A,B) [20].

We expected to observe decreasing atherosclerosis burden in the CL-treated LDLR-deficient mice, as reductions in adiposity and plasma lipids levels after dietary intervention have been shown to decrease lesion size [24,31], partially by promoting emigration of macrophages from lesions [42,43,44]. Despite an increased expression of *Cyp7a1* and reduced lipid levels in the CL–treated LDLR-deficient mice, we could not observe regression of atherosclerosis, unlike others have shown before with a simple dietary intervention switching from high-cholesterol diet to chow that led to improved systemic plasma glucose and lipid parameters [31,45,46,47]. One explanation for this finding could be the choice of the mouse model of atherosclerosis. As described previously [3] the effect of thermogenic BAT activation on atherosclerosis development depends on the mouse model used. In contrast to the mouse models used by Dong et al. [20], which have dysfunctional hepatic clearance of remnant particles [32,48,49], in E3L.CETP-mice the hepatic clearance route for remnant particles is only decelerated but still intact [50]. Thus, in the E3L.CETP-mice upon BAT activation, emerging cholesterol rich-remnant particles are still cleared from the circulation by the liver which subsequently protects from atherosclerosis [3]. As LDLR-deficient mice also still have functional APOE and regression of atherosclerosis in these mice has occurred solely by dietary intervention [31], we speculated that this model would be suitable to investigate regression of atherosclerosis upon activation of thermogenic adipocytes. However, although we observe slight reductions in plasma lipids most likely due to clearance by hepatic LDLR-related protein 1 (LRP1), the lack of hepatic APOE-LDLR route for clearance of remnant particles probably prevents better outcomes on the regression of atherosclerosis by CL treatment. In addition, more profound effects on plasma cholesterol lowering by CL treatment may be masked due to the dietary switch to chow diet, which by itself already profoundly lowered plasma lipid levels independently of CL. A CL intervention in this model without any dietary switch would most likely also result in unchanged atherosclerotic burden, similar to what has been observed previously [3]. In the end, as thermogenic activation with CL does not lead to any benefit neither on development nor regression of atherosclerosis, BAT activation doesn’t seem to benefit conditions that mimics human familial hypercholesterolemia. In the future, we plan to study plaque regression in the E3L.CETP-mice model, as due to the introduction of the CETP their lipoprotein metabolism is closer to the general human situation and thus makes the results readily translatable into humans beyond familial hypercholesterolemia. Indeed, cold exposure of humans rapidly increases small HDL particles with increased ABCA1-mediated cholesterol efflux [51], which points to an important role of CETP in this context.

## 5. Conclusions

In conclusion, although dietary CL treatment of LDLR-deficient mice with established atherosclerosis reduced adiposity and plasma lipid concentrations, these changes did not translate into reduced plaque burden in our experimental regimen. Future studies are required to explore the therapeutic potential of thermogenic adipocytes for the treatment of advanced atherosclerosis in a more refined manner.

## Figures and Tables

**Figure 1 nutrients-11-00463-f001:**
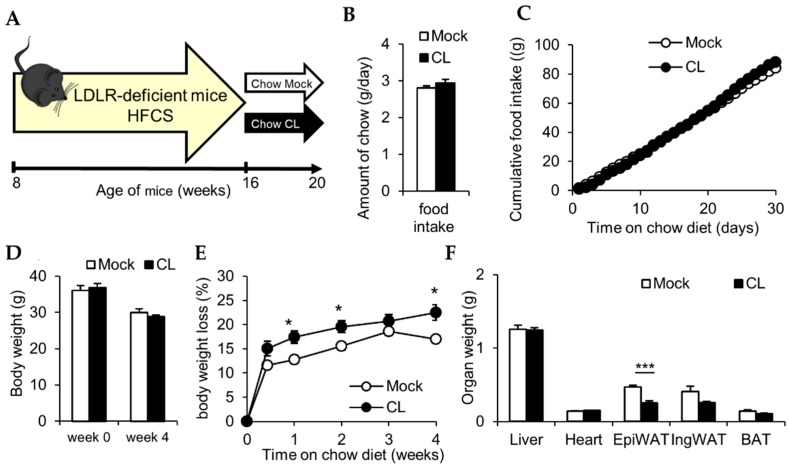
CL316,243 (CL) treatment reduces adiposity independently of food intake. (**A**) Study design: low-density lipoprotein receptor (LDLR)-deficient mice were fed a high-fat, high-cholesterol, and high-sucrose-diet (HFCS) for 8 weeks before they were switched to a chow diet supplemented with or without CL316,243 (CL) for 4 weeks; (**B**) Daily and (**C**) cumulative food intake (*n* = 7–9); (**D**) body weight before (week 0) and after (week 4) intervention with chow or chow + CL; (**E**) time course of body weight loss; (**F**) and organ weights after 4 weeks of chow (mock) or chow + CL (CL) feeding in 22 °C housed LDLR-deficient mice.

**Figure 2 nutrients-11-00463-f002:**
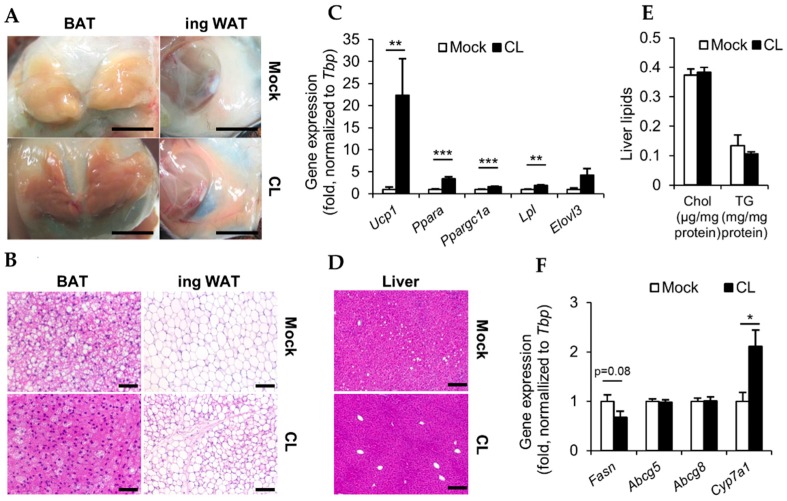
Effects of CL treatment on adipose tissues and liver; (**A**) representative macroscopic pictures of brown adipose tissue (BAT) and ingWAT (scale bar = 1 cm); (**B**) representative pictures of H&E-stained sections of BAT (scale bar = 50 µm) and ingWAT (scale bar = 100 µm); (**C**) relative gene expression normalized to *Tbp* as housekeeper in ingWAT (*n* = 7–9); (**D**) representative pictures of H&E-stained sections of livers (scale bar = 200 µm); (**E**) liver lipids (*n* = 7–9), and (**F**) relative gene expression normalized to *Tbp* as housekeeper in livers (*n* = 7–9) of LDLR-deficient mice fed a chow (mock) or chow + CL (CL) diet at 22 °C for 4 weeks.

**Figure 3 nutrients-11-00463-f003:**
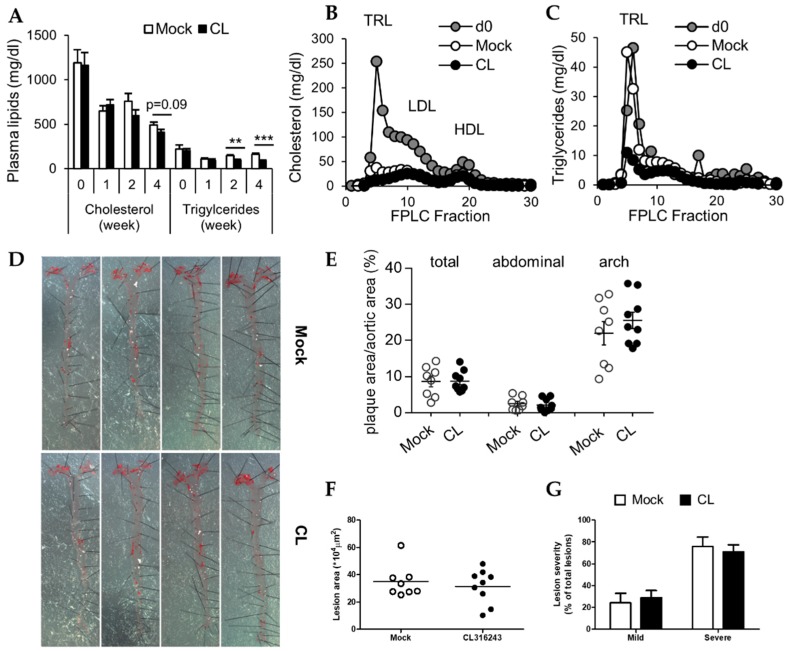
CL treatment reduces plasma lipid levels but does not affect atherosclerosis (**A**) Plasma cholesterol and triglyceride levels over 4 weeks chow or chow+ CL feeding; and (**B**) cholesterol and (**C**) triglyceride levels in fractionized plasma subjected to fast performance liquid chromatography (FPLC) before (d0) and after 4-week chow or chow + CL feeding in LDLR-deficient mice (*n* = 7–9). (**D**) Representative pictures of en face stained aortae with (**E**) corresponding quantification, as well as (**F**) quantification of lesion area and (**G**) and lesion severity of aortic roots LDLR-deficient mice fed a chow (mock) or chow + CL (CL) diet at 22 °C for 4 weeks (*n* = 7–9).

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
