# Peer review of "Effects of Pharmacological Thermogenic Adipocyte Activation on Metabolism and Atherosclerotic Plaque Regression"

_nutrients, 2019, doi:10.3390/nu11020463_

Round 1
Reviewer 1 Report
Worthmann A et al. reported that activation of adipose thermogenesis by β3AR-selective agonist (CL) significantly decreased body weight and plasma TG levels in LDL receptor (LDLR) KO mice. Despite the apparent effects of CL on adiposity, atherosclerotic plaque burden remained unchanged. The results are interesting; however, I have several comments to be addressed.
1. Introduction: Lack of the use of wild type control mice is a major limitation of this work. The authors should come clear why they did not use wild type control mice, and what is the rational reason of the use LDLR KO mice. The authors described their hypothesis on LDLR KO mice in the Discussion section (line 297-302) but such things should be firstly described in the Introduction.
2. Discussion: Because of the limitation said in comment 1, the authors should not conclude that “BAT activation doesn’t seem to benefit condition that mimics human familial hypercholesterolemia” (lines 309-310). To conclude this, the authors should use wild type mice. As the authors used only LDLR KO mice, but not wild type control, it is not clear whether 1) BAT activation does not lead any beneficial effect on hypercholesterolemia, or 2) LDLR is required for beneficial effect of CL on hypercholesterolemia. To minimize misunderstanding for the readers, the interpretations should be made more carefully. The sentence described in the Abstract “despite improving adiposity and plasma TG, pharmacologic activation of thermogenic adipocytes is not able to reverse atherosclerosis “in LDLR-deficient mice”” is reasonable.
3. Line 165: Figure 1D à Figure 1E
4. Line 169: Figure 1D à Figure 1F
Author Response
Worthmann A et al. reported that activation of adipose thermogenesis by β3AR-selective agonist (CL) significantly decreased body weight and plasma TG levels in LDL receptor (LDLR) KO mice. Despite the apparent effects of CL on adiposity, atherosclerotic plaque burden remained unchanged. The results are interesting; however, I have several comments to be addressed.
RESPONSE: We would like to thank the reviewer for her/his positive feedback and constructive criticism.
1. Introduction: Lack of the use of wild type control mice is a major limitation of this work. The authors should come clear why they did not use wild type control mice, and what is the rational reason of the use LDLR KO mice. The authors described their hypothesis on LDLR KO mice in the Discussion section (line 297-302) but such things should be firstly described in the Introduction.
RESPONSE: We appreciate the reviewer’s suggestions of using wild-type mice. However, in the introduction, we carefully explain the rationale for using the LDLR KO model, as wild-type mice do not develop atherosclerosis and therefore cannot be used to study the effect of BAT activation on atherosclerosis regression. To make this clearer we have further clarified this in the introduction:
Line 66: “While wild-type mice do not develop atherosclerosis, over the last decades the development of transgenic technology has made it possible to study atherosclerosis in preclinical mouse models.”
2. Discussion: Because of the limitation said in comment 1, the authors should not conclude that “BAT activation doesn’t seem to benefit condition that mimics human familial hypercholesterolemia” (lines 309-310). To conclude this, the authors should use wild type mice. As the authors used only LDLR KO mice, but not wild type control, it is not clear whether 1) BAT activation does not lead any beneficial effect on hypercholesterolemia, or 2) LDLR is required for beneficial effect of CL on hypercholesterolemia. To minimize misunderstanding for the readers, the interpretations should be made more carefully. The sentence described in the Abstract “despite improving adiposity and plasma TG, pharmacologic activation of thermogenic adipocytes is not able to reverse atherosclerosis “in LDLR-deficient mice”” is reasonable.
RESPONSE: We are thankful for the reviewer’s constructive criticism. We have previously shown that both cold and pharmacological BAT activation with CL316243 reduce plasma lipids in wild-type mice (Bartelt et al. Nature Medicine 2011, Bartelt et al. Nature Communications 2017) but also in other models with dyslipidemia like ApoE KO, LDLR KO and E3*Leiden.CETP mice (Berbee et al. Nature Communications 2015). However, as wild-type mice do not develop atherosclerosis and therefore cannot be used to study the effect of BAT activation on atherosclerosis regression, we did not include them in this study. To make this clearer we have further clarified this in the introduction.
Line 266: “While wild-type mice respond to the plasma lipid lowering effects of cold or CL, they do not develop atherosclerosis, and therefore transgenic models have to be used to study this process.”
3. Line 165: Figure 1D à Figure 1E
RESPONSE: We have corrected the error.
4. Line 169: Figure 1D à Figure 1F
RESPONSE: We have corrected the error.
Reviewer 2 Report
Comments to the manuscript nutrients-449548
The manuscript entitled “Effects of pharmacological thermogenic adipocyte activation on metabolism and atherosclerotic plaque regression” by Worthmann et al. investigates the impact of thermogenic adipose tissues activation on plaque regression in a mouse model of atherosclerosis. Thermogenic adipocytes in LDL receptor deficient mice were activated by dietary CL316,243 treatment and the outcomes on metabolically actve tissues, plasma lipids and atherosclerosis were analysed. While the chronic stimulation of thermogenic adipocytes reduced adiposity, increased WAT browning, altered liver gene expression and reduced plasma lipids levels, atherosclerotic plaque burden remained unchanged. The data presented suggest that despite improving adiposity and plasma TGs, the pharmacological activation of thermogenic adipocyte is not able to reverse atherosclerosis in LDLR-deficient mice.
The manuscript is well written and easy to follow; however, there are a couple of experiment that could considered to strength the quality of the manuscript.
Major comments
Results
Fig.2C. The authors should show also similar gene expression of BAT. Morover they should provide oxygen consumption and energy expenditure data of the CL vs control mice.
The authors should provide data of a similar experiment of CL-dietary challenge keeping the mice in HFCS diet and not going back to chow diet. That could potentially give a different outcome that the one obtained taking back the mice in chow diet when challenged with CL.
Author Response
The manuscript entitled “Effects of pharmacological thermogenic adipocyte activation on metabolism and atherosclerotic plaque regression” by Worthmann et al. investigates the impact of thermogenic adipose tissues activation on plaque regression in a mouse model of atherosclerosis. Thermogenic adipocytes in LDL receptor deficient mice were activated by dietary CL316,243 treatment and the outcomes on metabolically active tissues, plasma lipids and atherosclerosis were analysed. While the chronic stimulation of thermogenic adipocytes reduced adiposity, increased WAT browning, altered liver gene expression and reduced plasma lipids levels, atherosclerotic plaque burden remained unchanged. The data presented suggest that despite improving adiposity and plasma TGs, the pharmacological activation of thermogenic adipocyte is not able to reverse atherosclerosis in LDLR-deficient mice.
The manuscript is well written and easy to follow; however, there are a couple of experiment that could considered to strength the quality of the manuscript.
RESPONSE: We would like to thank the reviewer for her/his positive feedback and experimental suggestions.
Major comments
Results
1. Fig.2C. The authors should show also similar gene expression of BAT. Moreover, they should provide oxygen consumption and energy expenditure data of the CL vs control mice.
RESPONSE: We agree with the reviewer that these are standard benchmarking measurements to assure that the CL treatment is effective. However, we did not perform indirect calorimetry, as the CL effects are very well-established, and we have long-standing experience using this as a pharmacological approach to activate thermogenic adipocytes (e.g. Berbee et al Nature Communications 2015). Based on the brown fat macroscopic view and histology as well as gene expression/histology of the inguinal fat pad, the CL treatment is working effectively also in this study. Furthermore, the time frame for revising the paper does not allow repeating the study to perform the indirect calorimetry as suggested by the reviewer.
2. The authors should provide data of a similar experiment of CL-dietary challenge keeping the mice in HFCS diet and not going back to chow diet. That could potentially give a different outcome that the one obtained taking back the mice in chow diet when challenged with CL.
RESPONSE: We are thankful for this remark but essentially this experiment has been performed previously. As shown by Berbee et al Nature Communications 2015, LDLR KO mice continuously on a similar HFCS diet with CL do not show differences in atherogenesis. Hence, we decided to use the dietary switch to chow to stimulate atheroregression and test whether under these conditions, thermogenic adipocytes may have any beneficial effect. However, we acknowledged the reviewer’s comment by modifying the discussion:
Line 313: “A CL intervention in this model without any dietary switch would most likely also result in unchanged atherosclerotic burden, similar to what has been observed previously [3].”
Round 2
Reviewer 1 Report
The authors have revised the manuscript and answered adequately to my comments. I have no further comment.
Reviewer 2 Report
The way the authors addressed our comments is satisfactory.